# Trends in Transportation Modes and Time among Chinese Population from 2002 to 2012

**DOI:** 10.3390/ijerph17030945

**Published:** 2020-02-04

**Authors:** Weiyan Gong, Fan Yuan, Ganyu Feng, Yanning Ma, Yan Zhang, Caicui Ding, Zheng Chen, Ailing Liu

**Affiliations:** National Institute for Nutrition and Health, Chinese Center for Disease Control and Prevention, Beijing 100050, China; gongwy@ninh.chinacdc.cn (W.G.); yuanfan@ninh.chinacdc.cn (F.Y.); fenggy@ninh.chinacdc.cn (G.F.); mayn@ninh.chinacdc.cn (Y.M.); zhangyan@ninh.chinacdc.cn (Y.Z.); dingcc@ninh.chinacdc.cn (C.D.); chenzheng@ninh.chinacdc.cn (Z.C.)

**Keywords:** transportation mode, transportation time, trend, socioeconomic characteristics, China

## Abstract

Giving the rising trends in obesity and chronic diseases in China, strategies to increase physical activity are important. Transport-related activity is a substantial source of physical activity and can be easily incorporated into the daily lives. It is a key social factor of health, which can help improve people’s mental and physical health and decrease environmental pollution. However, little is known about recent trends in transportation modes and time in China. Between 2002 and 2010–2012, information about transportation behaviors of Chinese population aged 15 years or older were collected within two Chinese Nutrition and Health Surveys. A stratified multistage cluster sampling method was conducted to select participants. Sociodemographic information, transportation modes, and time were collected during face-to-face interviews. The study included 82,377 (mean age 41.2 years) and 143,075 (mean age 48.7 years) participants in the 2002 and 2010–2012 surveys respectively. The weighted prevalence of active transportation (including walking and cycling) decreased from 83.8% in 2002 to 54.3% in 2010–2012 (*p* < 0.001). During the same period, the number of participants using public transportation (including taking the bus, subway, or shuttle bus) has doubled (7.5% and 15.7%, respectively, for 2002 and 2010–2012, (*p* < 0.001)), and the proportion of inactive transportation (including driving or taking a car, motorcycle, taxi, or electric bicycle) more than tripled. Concurrently, the transportation time almost doubled with an increase of 25.9 min over the 10 years (*p* < 0.001). The prevalence of active transportation increased with age. Participants with higher family income and education reported a lower prevalence of active transportation. Females were more likely to use active transportation (OR (95% CI): 4.41 (4.14–4.70) and 2.50 (2.44–2.57), respectively, for 2002 and 2010–2012, where males were the reference). Before the prevalence of active transportation and physical activity gets lower, there is a need for the public health sector and the transport and land use sector to work together to develop related policies and initiatives with the aim of promoting active transportation and public transportation to increase the levels of physical activity and to decrease the risks of chronic diseases.

## 1. Introduction

China is facing an obesity and chronic diseases epidemic due to the population aging, urbanization, and unhealthy lifestyles [1]. Chronic diseases have become the leading cause of death among the Chinese population [2]. Insufficient physical activity is an important risk factor for obesity, cardiovascular disease, and diabetes [3,4,5,6,7]. Daily physical activities are most beneficial to the health of people, among which walking and cycling are the easiest to engage in and the most durable forms of physical activity. Physical activity often includes four domains based on the purpose or the location of activity: occupational activity, leisure exercise, domestic work, and transport-related activity. Transport-related activity is a substantial source of physical activity and can be easily incorporated into the daily lives. It has also become a focus for public health [8,9]. Studies have shown that active transportation (including walking and cycling) is associated with health benefits [10,11], including improvement of cardiovascular health [12,13], weight loss, and maintenance of healthy weight [14,15,16,17]. Compared to car, taxi, or motorcycle transportation, cycling or walking to and from work has been associated with a 50% lower risk for type 2 diabetes [18]. People who engaged in active transportation can increase physical activity [19]. One research showed that increasing walking and bicycling or reducing car travel could save $10–$63 million in estimated disease-specific direct treatment costs and indirect lost productivity costs per year in Nashville, Tennessee, USA [20]. 

Insisting on walking, cycling, and other active modes of transportation is one of the main means to increase the total daily physical activity of the population to prevent the occurrence of chronic diseases and to improve well-being. In view of mounting evidence on the health benefits of walking and cycling, some countries encouraged residents to increase transportation-related physical activity (such as walking or cycling to and from work) to improve individual health and to reduce air pollution, carbon emissions, traffic dangers, and other harmful impacts of car use. In Canada, suitable point-of-choice prompts were designed for university students to promote active transportation [21]. In the United States, community infrastructures were developed to support physical activity, such as sidewalks and bicycle trails [22]. An integrated transport and health impact model promoted population-level health impacts of shifting the population towards active transportation [23].

An increase in physical activity among the Chinese population is crucial because only 13.8% of them engage in physical activity and 42.0% of adults are overweight or obese [24,25]. Compared to 2002, the proportion of engaging in physical activity significantly decreased while the proportion of obesity and chronic diseases sharply increased. The paces of overweight and obesity and of chronic diseases increase are rapid in China, and the proportion is expected to increase to 69.9% by 2025. Despite extensive research conducted on the role of transportation on health, little is known about recent trends in transportation modes and time in China. Understanding the transportation modes and time can help develop effective public health interventions. The objective of the study was to describe trends in transportation modes and time among the Chinese population aged 15 years or older from 2002 to 2012 and to describe the sociodemographic factors in choosing transportation modes.

## 2. Materials and Methods

### 2.1. Study Design

Data used in this study were from Chinese Nutrition and Health Survey (2002) and Chinese Nutrition and Health Surveillance (2010–2012). The two surveys were nationally representative cross-sectional studies covering all 31 provinces, autonomous regions, and municipalities directly under the central government of China (except Taiwan, Hong Kong, and Macao). A stratified multistage cluster sampling method was conducted in the two surveys to recruit participants. Both survey research protocols were approved by the Ethics Committee of National Institute for Nutrition and Health (2013–018). Written informed consent was obtained from each participant. Basic information registration questionnaire contents included general demographic information (such as age, living area, gender, education level, family income, and so on) for both surveys.

In the 2002 survey, the whole country was divided into six strata: metropolises, small and medium-sized cities, and categories one to four villages based on their socioeconomic status, geographic characteristics, and social development information. A stratified multistage cluster sampling method was conducted to recruit participants from 132 surveillance sites of six strata areas. The personal health questionnaire was used to collect transportation mode- and time-related information. In the 2010–2012 survey, the whole country was divided into four strata: metropolises, small and medium-sized cities, normal villages, and poor villages based on their socioeconomic status, geographic characteristics, and social development information. The physical activity questionnaire was used to collect transportation mode- and time-related information.

Both surveys used face-to-face interviews, and the interviews were conducted at the homes of the participants by a well-trained investigator. To ensure data reliability, quality control measures, and evaluation indexes were made at the national, provincial, and district levels, quality control was carried out in the earlier and later stages of and during the field surveys.

### 2.2. Participants and Transportation Behaviors

The present study combined two datasets: 2002 and 2010–2012 for participants aged 15 years or older. For the purposes of this article, data from 225,452 respondents was analyzed, out of whom 82,377 were from the 2002 survey and 143,075 were from the 2010–2012 survey. Age was grouped as a categorical variable: 15–29.9 years, 30–44.9 years, 45–55.9 years, and 60 years or older. Residence was assigned for all participants as two strata areas: urban areas and rural areas. Educational level was categorized into three levels: primary school or below, junior or senior high school, and college or above. The socioeconomic status was grouped as follows: low, middle, high, and no response.

The participants were asked the following: what is the main transportation mode when you traveled in the past three months? How many minutes did you spend on travel per day? Four response options for transportation modes were available: walking; cycling; bus, subway, or shuttle bus; and car, electric bicycle, motorcycle, or taxi. The participants could select only one type of transportation mode. According to the contribution to the health and physical activity, the transportation modes were classified as active transportation, public transportation, and inactive transportation. Active transportation included walking and cycling; public transportation included taking a bus, subway, or shuttle bus; and inactive transportation included driving or taking a car, motorcycle, taxi, or electric bicycle.

### 2.3. Statistical Analyses

Due to the differences in age, gender, and regional distribution of the Chinese population between 2002 and 2010–2012, the prevalence of transportation modes and mean level of transportation time were standardized by age, gender, and region based on the 2010 national population census data, thus providing nationally representatively estimates. A complex sampling weight was assigned to each participant according to the study design. All statistics were calculated using SAS9.3 (SAS Institute, Cary, NC, USA). The Proc Surveyfreq and Proc Surveymean procedures in SAS were used to analyze the proportion, mean, and 95% confident interval (95% CI). The transportation modes were presented as percentages, and transportation time was expressed as mean. The Chi-square test was performed for comparing the proportion of transportation modes between the two surveys and in different subgroups. Survey multinomial logistic regression was used to evaluate association between the sociodemographic characteristic factors and the transportation modes. The Kolmogorov–Smirnov test was used to determine the normality. Due to the abnormal distribution, the Mann–Whitney U-test was used to examine the differences between genders, region types, and the two surveys of transportation time. The Kruskal–Wallis Rank sum test was used to examine the differences in age groups, education levels, and annual average income per capita. A two-tailed *p*-value < 0.05 was regarded as statistically significant.

## 3. Results

### 3.1. Characteristics of the Paticipants

The sociodemographic characteristics among the participants are shown in Table 1. The final sample sizes were 82,337 in the 2002 survey and 143,075 in the 2010–2012 survey. The mean age was 48.7 years in the 2010–2012 survey, which was older than the 2002 survey. Participants living in the urban areas in the 2010–2012 survey were more than the 2002 survey. There was a significant increase of annual average income in 2010–2012 survey with the development of the economy.

### 3.2. Trends in Tranportation Modes

Table 2 presents the prevalence of participants’ transportation modes by surveys and sociodemographic characteristics. The prevalence of active transportation decreased from 83.8% in 2002 to 54.3% in 2010–2012 (*p* < 0.001). The trends in the prevalence of transportation modes over the 2002–2012 surveys were generally similar for participants of almost all subgroups. The greatest decline occurred during the 30–44.9 age group compared to the other age groups, with a decrease by 44.4%. The proportion of participants living in rural areas who use active transportation declined more than that of those living in urban areas. The prevalence of active transportation among males declined more than among females. The declines in active transportation decreased with the increasing annual average income per capita. The lowest decrease in prevalence of active transportation was observed among the participants reported attending only primary school or below education compared to their counterparts.

The prevalence of public transportation increased from 7.5% in 2002 to 15.7% in 2010–2012, and it increased in all subgroups. More increases were among those who were female, were older, lived in rural areas, had lower education levels, and had lower annual average income per capita (Table 2). 

From 2002 to 2012, the number of participants using inactive transportation increased from 8.7% to 29.9% and a similar trend was found in each subgroup. The participants who were female and lived in rural areas reported higher increases than their counterparts. The participants who were in the ≥60 age group, with only primary school or below education level, and with low annual average income per capita had the highest increases compared with their counterparts.

The specific proportion of participants who walked; cycled; used public transportation; or rode in a car, on an electric bike, on a motorbike, or in a taxi for the whole sample are shown in Figure 1. The proportion of walking to travel declined from 57.5% in 2002 to 39.8% in 2010–2012. The fall in percentage of cycling for travel was more pronounced than walking, with a decline of 44.9%. Concurrently, a significant increase was observed for using public transportation. Meanwhile, the use of cars, electric bikes, motorbikes, or taxies almost tripled. The increase of passive transportation was mainly due to an increased use of cars, electronic bikes, motorbikes, or taxies over time.

### 3.3. Association between Tranportation Modes and Sociodemographic Characteristics

We entered the sociodemographic characteristic factors into multinomial logistic regression model to evaluate the association with transportation modes. Table 3 summarized the factors associated with active transportation and public transportation. Using the inactive transportation as a reference, all included sociodemographic characteristic factors significantly affected the other two classes (*p* < 0.001). In both surveys, females were significantly more likely to use active transportation (OR (95% CI): 4.41 (4.14–4.70) and 2.50 (2.44–2.57), respectively, for 2002 and 2010–2012, where males were the reference) and public transportation (OR (95% CI): 3.68 (3.36–4.03) and 2.20 (2.12–2.28), respectively, for 2002 and 2010–2012, where males were the reference). Whereas participants with high annual average income per capita were less likely to use active transportation and public transportation. The prevalence of active transportation and public transportation increased with age. The higher the education level, the higher the prevalence of public transportation. Using active transportation was significantly more common in participants living in rural areas than those living in urban areas in 2002. Public transportation use was more pronounced in urban areas than rural areas for both surveys; however, the gap was sharply reduced in 2010–2012.

### 3.4. Trends in Transportation time

There was a significant increase of transportation time between 2002 and 2010–2012 (Table 4). The participants in the 2010–2012 survey spent 63.0 min on travel, nearly twice as much for transportation time as the participants in the 2002 survey. The trends in transportation time over the 2002–2012 surveys were generally similar for participants of all subgroups. The largest growth in transportation time was the participants aged 30–44.9 years compared to the other age groups, increasing by 82.5%. The transportation time increased 27.5 min among males, and the increase was higher than that of females. Although participants living in urban areas reported slightly more transportation time than those in rural areas in both surveys, the increase in transportation time of rural participants was higher than that of urban participants. The participants who reported only attending junior or senior high school had the highest increase in transportation time compared with their counterparts. With respect to family income, the participants with low annual average income per capita increased 28.8 min on travel, which was more than that of those with middle and high annual average income per capita.

## 4. Discussion

This study investigated trends in transportation modes and time based on large, nationally representative surveys among the Chinese population. The analysis indicated that, although active transportation was still very prominent, a significant downtrend in the prevalence of active transportation was observed between 2002 and 2010–2012. Somewhat more worrisome, the population using inactive transportation increased dramatically, more than tripling. The reported levels of active transportation were still high when compared to the USA and UK [26,27], but the trends are inconsistent with the USA [26]. The uptrend in active transportation in USA may be related to a number of strategies, such as the construction of cycle paths, car restriction, car-free days, and so on [22,28]. Similar strategies have been implemented in other countries [21,29], such as Canada, Australia, etc. 

China’s urbanization has accelerated since 1996, and the rapid urbanization stage was 2002–2012. In 2012, the urbanization rate was 52.57% [30,31]. The decrease in active transportation was influenced by the rapid urbanization. The fact that the decrease in active transportation occurred primarily in rural areas in our study supports this viewpoint. The study also showed that, with respect to socioeconomic status and education level, participants with higher family income and education reported a lower prevalence of active transportation. This may be due to the fact that, with increased family income and decreased private car prices, the number of private cars has rapidly increased in recent years and the private car is going from a luxury to a necessary commodity. In 2012, China has topped the sales rankings in car sales for 4 years [32]. These people are more inclined to travel by private car and motorcycle by reason of convenience, timeliness, and reliability. In addition, high-income people bought big houses in an area far from work places to improve living conditions. They used inactive transportation, such as private car, motorcycle, and taxi to make up for the negative impact caused by the long distances to and from the place of residence. Factors other than urbanization and home–work separation have been shown to contribute to the choice of transportation modes including air pollution, personal safety, facilities to assist active transportation, net residential density, intersection density, and so on [33,34,35,36,37,38]. 

This study also showed that the transportation time almost doubled with an increase of 25.9 min over the past 10 years. The increase in transportation time was also influenced by the rapid urbanization and home–work separation. Long transportation time was associated with reductions in health-related activities, including sleeping, physical activity, food preparation, and time eating with family [39]. Workers with long transportation times were more likely to be absent from work, to have work accidents or lower levels of concentration, to be unemployed, and so on [40].

From 2002 to 2010–2012, the proportion of public transportation increased from 7.5% to 15.7%; nevertheless, the increase was much lower than in the inactive transportation. Compared with the people who used private cars, public transportation users had an easier time achieving 10,000 steps due to the walking associated with public transportation [41]. A study conducted in Shanghai showed that females and short-distance private car users were more likely to switch to public transportation [42]. When adjusting for sporting activity, people who regularly cycled for transport were still negatively correlated with medical risk factors [43]. Therefore, many countries strongly support cycling. Bike share programs were implemented as alternatives to motor vehicle use in Europe, Asia, and America [44,45,46], which provided convenience, low cost, reduced carbon, more efficient travel, health benefits, and decreased risk of bicycle crashes [44,47,48]. Given the association between transportation and human health, it is imperative that public health issues are considered in designing urban and transportation systems. The public transit systems and nonmotorized transit systems (such as bicycle trails and sidewalks) should be improved before the prevalence of active transportation gets lower. Bus rapid transit (BRT) systems should be vigorously developed and should allow users to take bicycles onto the buses. Beyond that, the network of BRT systems and bicycle trails should be integrated, which will be convenient for people who use active transportation and public transportation together. 

The major strengths of this study were that the sample was large and was nationally representative, and therefore, the results can be generalized to the Chinese people and can contribute to scientific evidence for the national policy on transportation mode and time. The main lack of this study was that the result was self-reported, where recall may lead to bias.

## 5. Conclusions

Active transportation to travel among the Chinese decreased significantly from 2002 to 2012, while inactive transportation increased sharply. In rural areas, higher education level and annual average income per capita were negatively associated with active transportation in 2010–2012. These trends may represent a critical loss of physical activity for the Chinese and may explain part of the increase in obesity and chronic diseases in China. Therefore, there is a need for the public health sector and the transport and land use sector to work together to develop related policies and initiatives with the aim of promoting active transportation among the Chinese by reducing structural and environmental barriers, thus contributing to increasing the levels of physical activity and decreasing the risks of chronic diseases. 

## Figures and Tables

**Figure 1 ijerph-17-00945-f001:**
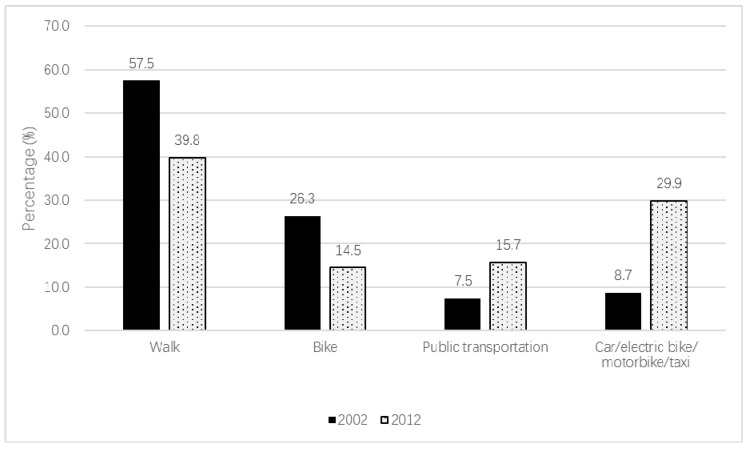
Specific proportion of participants who walked; cycled; used public transportation (taking a bus, subway, or shuttle bus); or rode in a car, on an electric bike, on a motorbike, or in a taxi to travel.

**Table 1 ijerph-17-00945-t001:** Sociodemographic characteristics of the participants between the years 2002 and 2010–2012.

Characteristics	Total	2002	2010–2012	*p*-Value
Total, *n* (%)	225,452	82,377	143,075	
Gender, *n* (%)				<0.001
Male	108,173 (48.0)	42,734 (51.9)	65,439 (45.7)	
Female	117,279 (52.0)	39,643 (48.1)	77,636 (54.3)	
Age (year), Mean ± SD	46.0 ± 16.2	41.2 ± 14.4	48.7 ± 16.6	<0.001
Age group (year), *n* (%)				<0.001
15–29.9	40,172 (17.8)	17,902 (21.7)	22,270 (15.6)	
30–44.9	64,784 (28.7)	31,231 (37.9)	33,553 (23.5)	
45–55.9	72,373 (32.1)	23,644 (28.7)	48,729 (34.1)	
≥60	48,123 (21.4)	9,600 (11.7)	38,523 (26.9)	
Region type, *n* (%)				<0.001
Urban areas	96,038 (42.6)	23,315 (28.3)	72,723 (50.8)	
Rural areas	129,414 (57.4)	59,062 (71.7)	70,352 (49.2)	
Education level, *n* (%)				<0.001
Primary school or below	88,772 (39.4)	34,222 (41.5)	54,550 (38.1)	
Junior or senior high school	120,036 (53.2)	43,180 (52.4)	76,856 (53.7)	
College and above	16,644 (7.4)	4975 (6.0)	11,669 (8.2)	
Annual average income per capita, *n* (%)				<0.001
Low	97,794 (43.4)	48,864 (59.3)	48,930 (34.2)	
Middle	64,237 (28.5)	21,147 (25.7)	43,090 (30.1)	
High	51,833 (23.0)	10,844 (13.2)	40,989 (28.7)	
No response	11,588 (5.1)	1522 (1.9)	10,066 (7.0)	

**Table 2 ijerph-17-00945-t002:** Prevalence and trends of transportation mode between the years 2002 and 2010–2012 ^#^.

Characteristics	2002 Transportation Mode (%, 95% CI) ^##^			2010–2012 Transportation Mode (%) ^##^			*p*-Value
	Active *	Public **	Inactive ***	Active *	Public **	Inactive ***	
Total	83.8 (83.5–84.1)	7.5 (7.3–7.8)	8.7 (8.4–8.9)	54.3 (54.0–54.6)	15.7 (15.5–16.0)	29.9 (29.6–30.2)	<0.001
Gender							
Male	79.2 (78.7–79.7)	7.5 (7.1–7.8)	13.3 (13.0–13.7)	46.7 (46.2–47.2)	14.9 (14.5–15.2)	38.4 (37.9–38.9)	<0.001
Female	88.6 (88.2–89.0)	7.6 (7.2–7.9)	3.8 (3.6–4.0)	62.2 (61.8–62.6)	16.6 (16.3–17.0)	21.2 (20.8–21.5)	<0.001
Age group							
15–29.9	76.8 (76.1–77.6)	12.5 (11.9–13.1)	10.7 (10.2–11.2)	46.1 (45.3–46.9)	23.3 (22.6–23.9)	30.6 (29.9–31.3)	<0.001
30–44.9	80.6 (80.1–81.2)	6.6 (6.2–6.9)	12.8 (12.4–13.2)	44.8 (44.3–45.4)	14.1 (13.7–14.4)	41.1 (40.6–41.6)	<0.001
45–55.9	88.7 (88.2–89.2)	5.6 (5.2–5.9)	5.8 (5.4–6.1)	60.1 (59.7–60.6)	12.2 (11.9–12.5)	27.7 (27.3–28.1)	<0.001
≥60	95.9 (95.4–96.4)	2.9 (2.5–3.3)	1.2 (0.9–1.5)	79.8 (79.4–80.2)	10.1 (9.8–10.4)	10.1 (9.8–10.4)	<0.001
Region type							
Urban areas	76.3 (75.7–76.9)	13.7 (13.3–14.2)	10.0 (9.6–10.4)	52.7 (52.3–53.2)	21.4 (21.1–21.8)	25.8 (25.4–26.2)	<0.001
Rural areas	91.3 (91.1–91.6)	1.3 (1.2–1.4)	7.3 (7.1–7.6)	55.9 (55.5–56.4)	10.1 (9.8–10.3)	34.0 (33.6–34.5)	<0.001
Education level							
Primary school or below	95.3 (95.0–95.5)	1.3 (1.1–1.5)	3.4 (3.2–3.6)	67.9 (67.5–68.4)	9.2 (8.9–9.5)	22.9 (22.4–23.3)	<0.001
Junior or senior high school	80.1 (79.6–80.6)	8.6 (8.2–8.9)	11.3 (11.0–11.7)	50.7 (50.2–51.1)	15.6 (15.3–15.9)	33.8 (33.4–34.2)	<0.001
College and above	62.4 (60.9–63.9)	24.8 (23.4–26.1)	12.8 (11.9–13.8)	37.3 (36.3–38.3)	33.6 (32.6–34.7)	29.1 (28.1–30.0)	<0.001
Annual average income per capita							
Low	90.0 (89.7–90.4)	4.4 (4.2–4.7)	5.5 (5.3–5.8)	62.4 (61.8–62.9)	13.1 (12.7–13.5)	24.5 (24.0–25.0)	<0.001
Middle	79.8 (79.1–80.5)	9.5 (9.0–10.1)	10.7 (10.2–11.2)	53.7 (53.1–54.2)	16.5 (16.0–16.9)	29.8 (29.3–30.4)	<0.001
High	68.8 (67.7–69.8)	15.0 (14.2–15.9)	16.2 (15.5–17.0)	46.4 (45.8–46.9)	16.9 (16.4–17.3)	36.8 (36.2–37.3)	<0.001
No response	76.9 (74.4–79.5)	10.9 (8.9–12.9)	12.1 (10.2–14.1)	52.2 (51.0–53.3)	20.3 (19.3–21.2)	27.6 (26.5–28.6)	<0.001

^#^ Standardized by age, gender, and region based on the 2010 national population. ^##^ Sample weighted results. * Active transportation: walking or cycling. ** Public transportation: taking a bus, subway, or shuttle bus. *** Inactive transportation: driving or taking a car, motorcycle, taxi, or electric bicycle. *p*-Value is the test result between two different survey.

**Table 3 ijerph-17-00945-t003:** Survey multinomial logistic regress analysis of factors associated with transportation modes ^#^.

Characteristics	2002 (OR (95%CI)) ^##^		2010–2012 (OR (95%CI)) ^##^	
	Active *	Public **	Active *	Public **
Gender				
Male	1.0	1.0	1.0	1.0
Female	4.41 (4.14–4.70)	3.68 (3.36–4.03)	2.50 (2.44–2.57)	2.20 (2.12–2.28)
Age group				
15–29.9	1.0	1.0	1.0	1.0
30–44.9	0.84 (0.79–0.89)	0.42 (0.38–0.47)	0.55 (0.53–0.58)	0.37 (0.35–0.39)
45–55.9	2.05 (1.89–2.22)	0.83 (0.74–0.93)	1.10 (1.06–1.14)	0.51 (0.48–0.53)
≥60	10.18 (8.35–12.42)	2.08 (1.62–2.68)	3.51 (3.34–3.69)	1.16 (1.09–1.23)
Region type				
Urban areas	1.0	1.0	1.0	1.0
Rural areas	1.56 (1.47–1.65)	0.16 (0.14–0.17)	0.68 (0.66–0.70)	0.41 (0.40–0.43)
Education level				
Primary school or below	1.0	1.0	1.0	1.0
Junior or senior high school	0.54 (0.50–0.58)	1.64 (1.42–1.90)	0.74 (0.72–0.77)	1.03 (0.99–1.08)
College and above	0.60 (0.53–0.67)	3.09 (2.58–3.71)	0.61 (0.57–0.64)	1.62 (1.51–1.73)
Annual average income per capita				
Low	1.0	1.0	1.0	1.0
Middle	0.45 (0.42–0.48)	0.86 (0.77–0.95)	0.73 (0.71–0.76)	0.92 (0.88–0.96)
High	0.22 (0.20–0.23)	0.63 (0.57–0.71)	0.55 (0.53–0.57)	0.78 (0.75–0.82)
No response	0.38 (0.32–0.45)	0.73 (0.57–0.94)	0.77 (0.73–0.82)	1.00 (0.93–1.07)

^#^ Standardized by age, gender, and region based on the 2010 national population: inactive transportation was used as the class of reference. ^##^ Sample weighted results. * Active transportation: walking or cycling. ****** Public transportation: taking a bus, subway, or shuttle bus.

**Table 4 ijerph-17-00945-t004:** Trends in transportation time between 2002 and 2010–2012 ^#^.

Characteristics	2002 Transportation Time (min) ^##^		2010–2012 Transportation Time (min) ^##^		Trend *p*-Value
	Mean (SE)	*p*-Value	Mean (SE)	*p*-Value	
Total	37.1 (0.1)		63.0 (0.2)		<0.001 ^a^
Gender		0.233 ^a^		<0.001 ^a^	
Male	37.7 (0.2)		65.2 (0.3)		<0.001 ^a^
Female	36.5 (0.2)		60.8 (0.2)		<0.001 ^a^
Age group		<0.001 ^b^		<0.001 ^b^	
15–29.9	40.3 (0.3)		58.6 (0.4)		<0.001 ^a^
30–44.9	34.8 (0.2)		63.5 (0.3)		<0.001 ^a^
45–55.9	35.8 (0.2)		65.3 (0.2)		<0.001 ^a^
≥60	37.4 (0.4)		67.1 (0.3)		<0.001 ^a^
Region type		<0.001 ^a^		<0.001 ^a^	
Urban areas	40.0 (0.2)		65.8 (0.2)		<0.001 ^a^
Rural areas	34.2 (0.1)		60.3 (0.2)		<0.001 ^a^
Education level		<0.001 ^b^		<0.001 ^b^	
Primary school or below	35.8 (0.2)		64.3 (0.3)		<0.001 ^a^
Junior or senior high school	36.8 (0.2)		61.6 (0.2)		<0.001 ^a^
College and above	43.5 (0.6)		66.7 (0.5)		<0.001 ^a^
Annual average income per capita		<0.001 ^b^		<0.001 ^b^	
Low	36.5 (0.2)		65.3 (0.3)		<0.001 ^a^
Middle	36.5 (0.3)		61.7 (0.3)		<0.001 ^a^
High	40.1 (0.4)		62.4 (0.3)		<0.001 ^a^
No response	38.1 (1.0)		60.6 (0.6)		<0.001 ^a^

^#^ Standardized by age, gender, and region based on the 2010 national population. ^##^ Sample weighted results. ^a^ Mann–Whitney U test. ^b^ Kruskal–Wallis Rank sum test. *p*-Value is the test result of different subgroups among each survey; Trend *p* Value is the test result between two different survey.

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
