# Peer review of "Trends in Transportation Modes and Time among Chinese Population from 2002 to 2012"

_ijerph, 2020, doi:10.3390/ijerph17030945_

Round 1

Reviewer 1 Report

Dear authors, The paper is very great.

I suggest to adjust the data for gender. I suggest to adjust the transportation modes by gender (using a logistic regression analysis). Indeed, it is a variable with could affect the association between transportation modes and metabolic syndrome.

Author Response

Dear reviewer,

We are extremely grateful for your positive and constructive feedback. The incorporation of your suggestions has strengthened the manuscript. According to your advice, we revised the relevant part, and highlighted that in manuscript. Here below is our description on revision, we attached the Word file too.

If you have any question, please contact us without hesitate.

Thank you for the kind advices.

Sincerely yours,

Weiyan Gong

Description on revision:

Comment: I suggest to adjust the data for gender. I suggest to adjust the transportation modes by gender (using a logistic regression analysis). Indeed, it is a variable with could affect the association between transportation modes and metabolic syndrome.

Response: According to your suggestion, we added logistic regression analysis and adjusted the transportation modes by socio-demographic characteristics including gender. We revised the content as follows:

(1)Abstract (page 1 line 28-31):

The prevalence of active transportation increased with age. Participants with higher family income and education reported a lower prevalence of active transportation. Female was more likely to use active transportation (OR (95%CI): 4.41(4.14-4.70), 2.50(2.44-2.57), respectively for 2002 and 2010-2012, where male was the reference).

(2)2.3. Statistical Analyses (page 3 line 128-129)

Survey multinomial logistic regression was used to evaluate association between the socio-demographic characteristic factors and the transportation modes.

(3)3.3. Association between Tranportation Modes and Socio-demographic Characteristics (page 6,7 line 180-197)

We entered socio-demographic characteristic factors into multinomial logistic regression model to evaluate the association with transportation modes. Table 3 summarized the factors associated with active transportation and public transportation. Using the inactive transportation as a reference, all included socio-demographic characteristic factors significantly affected the other two classes (p<0.001). In both surveys, female was significantly more likely to use active transportation (OR (95% CI): 4.41(4.14-4.70), 2.50(2.44-2.57) respectively for 2002 and 2010-2012, where the male was reference) and public transportation (OR (95% CI): 3.68(3.36-4.03), 2.20(2.12-2.28) respectively for 2002 and 2010-2012, where the male was reference). Whereas participants with high annual average income per capita were less likely to use active transportation and public transportation. The prevalence of active transportation and public transportation increased with the age. The higher the education level the higher the prevalence of public transportation. Using active transportation was significantly more common in participants living in rural areas than those living in urban areas in 2002. Public transportation use was more pronounced in urban areas than rural areas for both surveys, however, the gap was sharply reduced in 2010-2012.

Table 3. Survey multinomial logistic regress analysis of factors associated with transportation modes #

Characteristics

2002 (OR(95%CI))##

2010-2012(OR(95%CI)) ##

Active*

Public**

Active*

Public**

Gender

Male

1.0

1.0

1.0

1.0

Female

4.41(4.14-4.70)

3.68(3.36-4.03)

2.50(2.44-2.57)

2.20(2.12-2.28)

Age group

15-29.9

1.0

1.0

1.0

1.0

30-44.9

0.84(0.79-0.89)

0.42(0.38-0.47)

0.55(0.53-0.58)

0.37(0.35-0.39)

45-55.9

2.05(1.89-2.22)

0.83(0.74-0.93)

1.10(1.06-1.14)

0.51(0.48-0.53)

≥60

10.18(8.35-12.42)

2.08(1.62-2.68)

3.51(3.34-3.69)

1.16(1.09-1.23)

Region type

Urban areas

1.0

1.0

1.0

1.0

Rural areas

1.56(1.47-1.65)

0.16(0.14-0.17)

0.68(0.66-0.70)

0.41(0.40-0.43)

Education level

Primary school or below

1.0

1.0

1.0

1.0

Junior or Senior high school

0.54(0.50-0.58)

1.64(1.42-1.90)

0.74(0.72-0.77)

1.03(0.99-1.08)

College and above

0.60(0.53-0.67)

3.09(2.58-3.71)

0.61(0.57-0.64)

1.62(1.51-1.73)

Annual average income per capita

Low

1.0

1.0

1.0

1.0

Middle

0.45(0.42-0.48)

0.86(0.77-0.95)

0.73(0.71-0.76)

0.92(0.88-0.96)

High

0.22(0.20-0.23)

0.63(0.57-0.71)

0.55(0.53-0.57)

0.78(0.75-0.82)

No response

0.38(0.32-0.45)

0.73(0.57-0.94)

0.77(0.73-0.82)

1.00(0.93-1.07)

# standardized by age, gender and region based on the 2010 national population, inactive transportation was used as class of reference; ## sample weighted results; * Active transportation: walking or cycling; ** Public transportation: taking a bus/subway/shuttle bus.

(4)5.Conclusions (page 9 line 268,269)

In rural area, with higher education level and annual average income per capita were negatively with active transportation in 2010-2012.

Reviewer 2 Report

The entire manuscript should be edited for the English language and style. There are many missing words, incorrect words (e.g., "per capital" should be "per capita", and incomplete thoughts (e.g., pg 2, line 58/59). 

When deciding transportation behaviors to be active or inactive - why was public transit denoted to be "inactive" when in the discussion section, a point is made for public transit in the increase of physical activity (i.e., line 207/208).

Author Response

Dear reviewer,

We are extremely grateful for your positive and constructive feedback. The incorporation of your suggestions has strengthened the manuscript. According to your advice, we revised the relevant part, and highlighted that in manuscript. Here below is our description on revision, we attached the Word file too. 

If you have any question, please contact us without hesitate.

Thank you for the kind advices.

Sincerely yours,

Weiyan Gong

Description on revision:

Comment 1: The entire manuscript should be edited for the English language and style. There are many missing words, incorrect words (e.g., "per capital" should be "per capita", and incomplete thoughts (e.g., pg 2, line 58/59). 

Response: We have added the missing words, revised the incorrect words and edited the manuscript (page 2 line 63, 76, 92;page 3 line 97,98, 127,133; page 4 line 142,152; page 7 line 209,210; page 8 line 235)

Comment 2:When deciding transportation behaviors to be active or inactive - why was public transit denoted to be "inactive" when in the discussion section, a point is made for public transit in the increase of physical activity (i.e., line 207/208).

Response: It is true that public transit increased physical activity when compared to motorized transit (car, motorcycle, taxi, etc.), and it should not be inactive transportation. So we classified the transportation modes as active, public and inactive according to your suggestion. We revised the content as follows:

(1)Abstract(page 1 line 23-27):

During the same period, the number of participants using public transportation (including taking a bus/subway/shuttle bus) has doubled (7.5%, 15.7% respectively for 2002 and 2010-2012, (p<0.001), and the proportion of inactive transportation (including driving or taking a car/motorcycle/taxi/an electric bicycle) more than tripled.

(2)2.2. Participants and Transportation Behaviors(page 3 line 113-117):

According to the contribution to the health and physical activity, the transportation modes were classified as active transportation, public transportation and inactive transportation. The active transportation included walking and cycling, the public transportation included taking a bus/subway/shuttle bus, and the inactive transportation included driving or taking a car/motorcycle/taxi/an electric bicycle.

(3)3.2. Trends in Tranportation Modes(page 4,5 line 155-167):

The prevalence of public transportation increased from 7.5% in 2002 to 15.7% in 2010-2012, and it increased in all subgroups. More increases were among those who were female, older, in rural area, lower education levels and lower annual average income per capita (Table 2).

From 2002 to 2012, the number of participants use inactive transportation increased from 8.7% to 29.9%, and the similar trend was found in each subgroup. The participants who were female and in rural area reported higher increase than their counterparts. The participants who were in ≥60 age group, with primary school or below education level, and with low annual average income per capita had the highest increase compared with their counterparts.      

Table 2. Prevalence and trends of transportation mode between the years 2002 and 2010-2012#

Characteristics

2002 Transportation Mode (%, 95% CI)##

2010-2012 Transportation Mode (%)##

p Value

Active*

Public**

Inactive***

Active*

Public**

Inactive***

Total

83.8(83.5-84.1)

7.5(7.3-7.8)

8.7(8.4-8.9)

54.3(54.0-54.6)

15.7(15.5-16.0)

29.9(29.6-30.2)

<0.001

Gender

Male

79.2(78.7-79.7)

7.5(7.1-7.8)

13.3(13.0-13.7)

46.7(46.2-47.2)

14.9(14.5-15.2)

38.4(37.9-38.9)

<0.001

Female

88.6(88.2-89.0)

7.6(7.2-7.9)

3.8(3.6-4.0)

62.2(61.8-62.6)

16.6(16.3-17.0)

21.2(20.8-21.5)

<0.001

Age group

15-29.9

76.8(76.1-77.6)

12.5(11.9-13.1)

10.7(10.2-11.2)

46.1(45.3-46.9)

23.3(22.6-23.9)

30.6(29.9-31.3)

<0.001

30-44.9

80.6(80.1-81.2)

6.6(6.2-6.9)

12.8(12.4-13.2)

44.8(44.3-45.4)

14.1(13.7-14.4)

41.1(40.6-41.6)

<0.001

45-55.9

88.7(88.2-89.2)

5.6(5.2-5.9)

5.8(5.4-6.1)

60.1(59.7-60.6)

12.2(11.9-12.5)

27.7(27.3-28.1)

<0.001

≥60

95.9(95.4-96.4)

2.9(2.5-3.3)

1.2(0.9-1.5)

79.8(79.4-80.2)

10.1(9.8-10.4)

10.1(9.8-10.4)

<0.001

Region type

Urban areas

76.3(75.7-76.9)

13.7(13.3-14.2)

10.0(9.6-10.4)

52.7(52.3-53.2)

21.4(21.1-21.8)

25.8(25.4-26.2)

<0.001

Rural areas

91.3(91.1-91.6)

1.3(1.2-1.4)

7.3(7.1-7.6)

55.9(55.5-56.4)

10.1(9.8-10.3)

34.0(33.6-34.5)

<0.001

Education level

Primary school or below

95.3(95.0-95.5)

1.3(1.1-1.5)

3.4(3.2-3.6)

67.9(67.5-68.4)

9.2(8.9-9.5)

22.9(22.4-23.3)

<0.001

Junior or Senior high school

80.1(79.6-80.6)

8.6(8.2-8.9)

11.3(11.0-11.7)

50.7(50.2-51.1)

15.6(15.3-15.9)

33.8(33.4-34.2)

<0.001

College and above

62.4(60.9-63.9)

24.8(23.4-26.1)

12.8(11.9-13.8)

37.3(36.3-38.3)

33.6(32.6-34.7)

29.1(28.1-30.0)

<0.001

Annual average income per capita

Low

90.0(89.7-90.4)

4.4(4.2-4.7)

5.5(5.3-5.8)

62.4(61.8-62.9)

13.1(12.7-13.5)

24.5(24.0-25.0)

<0.001

Middle

79.8(79.1-80.5)

9.5(9.0-10.1)

10.7(10.2-11.2)

53.7(53.1-54.2)

16.5(16.0-16.9)

29.8(29.3-30.4)

<0.001

High

68.8(67.7-69.8)

15.0(14.2-15.9)

16.2(15.5-17.0)

46.4(45.8-46.9)

16.9(16.4-17.3)

36.8(36.2-37.3)

<0.001

No response

76.9(74.4-79.5)

10.9(8.9-12.9)

12.1(10.2-14.1)

52.2(51.0-53.3)

20.3(19.3-21.2)

27.6(26.5-28.6)

<0.001

# standardized by age, gender and region based on the 2010 national population; ## sample weighted results; * Active transportation: walking or cycling; ** Public transportation: taking a bus/subway/shuttle bus; *** Inactive transportation: driving or taking a car/motorcycle/taxi/an electric bicycle; p Value is the test result between two different survey.

(4)4.Discussion(page 9 line 246-247 )

From 2002 to 2010-2012, the proportion of public transportation increased from 7.5% to 15.7%, nevertheless, the increase was much lower than the inactive transportation.

Reviewer 3 Report

The paper entitled “Trends in Transportation Modes and Time among 2 Chinese Population from 2002 to 2012” presents an interesting and relevant topic to public health, as it analyses temporal trends in active transportation modes in China. The paper is easy-to-read, generally well-written and justified, and the data seems robust for fulfilling their objectives. Thus, I think the paper can be publishable in the IJERPH. However, there are some issues that the authors need to consider for a further version:

My main concerns are:

It is not clear why 2002 and 2012 is the adequate time frame for this study. When did the rapid urbanization of China start? Perhaps more data on the history of Chinese changes in these years could make it more clear. The authors need to specify better the definitions of the variables, especially if they have been used in previous studies. My main worry is the classification of active vs non-active. Many studies include bus or train as a mid-active mode of transportation and not a passive mode of transportation. I´d suggest the authors include the use of public transportation as a different category The analysis plan might be too simple. A more complex analysis (e.g. regression analysis) could take into account the joint effects of the explanatory variables. I strongly suggest to consider alternative analysis.

Author Response

Dear reviewer,

We are extremely grateful for your positive and constructive feedback. The incorporation of your suggestions has strengthened the manuscript. According to your advice, we revised the relevant part, and highlighted that in manuscript. Here below is our description on revision, we attached the Word file too. 

If you have any question, please contact us without hesitate.

Thank you for the kind advices.

Sincerely yours,

Weiyan Gong

Description on revision:

Comment 1: It is not clear why 2002 and 2012 is the adequate time frame for this study. When did the rapid urbanization of China start? Perhaps more data on the history of Chinese changes in these years could make it more clear.

Response: According to your suggestion, we add the data on the history of Chinses changes in these years as follow: (page 8 line 225,226)

China’s urbanization has accelerated since 1996, and the rapid urbanization stage was 2002-2012. In 2012, the urbanization rate was 52.57% [31,32].

Comment 2:The authors need to specify better the definitions of the variables, especially if they have been used in previous studies. My main worry is the classification of active vs non-active. Many studies include bus or train as a mid-active mode of transportation and not a passive mode of transportation. I´d suggest the authors include the use of public transportation as a different category The analysis plan might be too simple. A more complex analysis (e.g. regression analysis) could take into account the joint effects of the explanatory variables. I strongly suggest to consider alternative analysis. 

Response: According to your suggestion, we reclassified the transportation modes as: active, public and inactive. We also added regression analysis. Please see the revised contents as follows:

(1)Abstract (page 1 line 23-31):

During the same period, the number of participants using public transportation (including taking a bus/subway/shuttle bus) has doubled (7.5%, 15.7% respectively for 2002 and 2010-2012, (p<0.001), and the proportion of inactive transportation (including driving or taking a car/motorcycle/taxi/an electric bicycle) more than tripled. Concurrently, the transportation time almost doubled with an increase of 25.9 minutes over the 10 years (p<0.001). The prevalence of active transportation increased with age. Participants with higher family income and education reported a lower prevalence of active transportation. Female was more likely to use active transportation (OR (95%CI): 4.41(4.14-4.70), 2.50(2.44-2.57), respectively for 2002 and 2010-2012, where male was the reference).

(2)2.2. Participants and Transportation Behaviors (page 3 line 113-117):

According to the contribution to the health and physical activity, the transportation modes were classified as active transportation, public transportation and inactive transportation. The active transportation included walking and cycling, the public transportation included taking a bus/subway/shuttle bus, and the inactive transportation included driving or taking a car/motorcycle/taxi/an electric bicycle.

(3)2.3. Statistical Analyses (page 3 line 128-129)

Survey multinomial logistic regression was used to evaluate association between the socio-demographic characteristic factors and the transportation modes.

(4)3.2. Trends in Tranportation Modes(page 4,5 line 155-167):

The prevalence of public transportation increased from 7.5% in 2002 to 15.7% in 2010-2012, and it decreased in all subgroups. More increases were among those who were female, older, in rural area, lower education levels and lower annual average income per capita (Table 2).

From 2002 to 2012, the number of participants use inactive transportation increased from 8.7% to 29.9%, and the similar trend was found in each subgroup. The participants who were female and in rural area reported higher increase than their counterparts. The participants who were in 60 age group, primary school or below education level, and low annual average income per capita had the highest increase compared with their counterparts.

Table 2. Prevalence and trends of transportation mode between the years 2002 and 2010-2012#

Characteristics

2002 Transportation Mode (%, 95% CI)##

2010-2012 Transportation Mode (%)##

p Value

Active*

Public**

Inactive***

Active*

Public**

Inactive***

Total

83.8(83.5-84.1)

7.5(7.3-7.8)

8.7(8.4-8.9)

54.3(54.0-54.6)

15.7(15.5-16.0)

29.9(29.6-30.2)

<0.001

Gender

Male

79.2(78.7-79.7)

7.5(7.1-7.8)

13.3(13.0-13.7)

46.7(46.2-47.2)

14.9(14.5-15.2)

38.4(37.9-38.9)

<0.001

Female

88.6(88.2-89.0)

7.6(7.2-7.9)

3.8(3.6-4.0)

62.2(61.8-62.6)

16.6(16.3-17.0)

21.2(20.8-21.5)

<0.001

Age group

15-29.9

76.8(76.1-77.6)

12.5(11.9-13.1)

10.7(10.2-11.2)

46.1(45.3-46.9)

23.3(22.6-23.9)

30.6(29.9-31.3)

<0.001

30-44.9

80.6(80.1-81.2)

6.6(6.2-6.9)

12.8(12.4-13.2)

44.8(44.3-45.4)

14.1(13.7-14.4)

41.1(40.6-41.6)

<0.001

45-55.9

88.7(88.2-89.2)

5.6(5.2-5.9)

5.8(5.4-6.1)

60.1(59.7-60.6)

12.2(11.9-12.5)

27.7(27.3-28.1)

<0.001

≥60

95.9(95.4-96.4)

2.9(2.5-3.3)

1.2(0.9-1.5)

79.8(79.4-80.2)

10.1(9.8-10.4)

10.1(9.8-10.4)

<0.001

Region type

Urban areas

76.3(75.7-76.9)

13.7(13.3-14.2)

10.0(9.6-10.4)

52.7(52.3-53.2)

21.4(21.1-21.8)

25.8(25.4-26.2)

<0.001

Rural areas

91.3(91.1-91.6)

1.3(1.2-1.4)

7.3(7.1-7.6)

55.9(55.5-56.4)

10.1(9.8-10.3)

34.0(33.6-34.5)

<0.001

Education level

Primary school or below

95.3(95.0-95.5)

1.3(1.1-1.5)

3.4(3.2-3.6)

67.9(67.5-68.4)

9.2(8.9-9.5)

22.9(22.4-23.3)

<0.001

Junior or Senior high school

80.1(79.6-80.6)

8.6(8.2-8.9)

11.3(11.0-11.7)

50.7(50.2-51.1)

15.6(15.3-15.9)

33.8(33.4-34.2)

<0.001

College and above

62.4(60.9-63.9)

24.8(23.4-26.1)

12.8(11.9-13.8)

37.3(36.3-38.3)

33.6(32.6-34.7)

29.1(28.1-30.0)

<0.001

Annual average income per capita

Low

90.0(89.7-90.4)

4.4(4.2-4.7)

5.5(5.3-5.8)

62.4(61.8-62.9)

13.1(12.7-13.5)

24.5(24.0-25.0)

<0.001

Middle

79.8(79.1-80.5)

9.5(9.0-10.1)

10.7(10.2-11.2)

53.7(53.1-54.2)

16.5(16.0-16.9)

29.8(29.3-30.4)

<0.001

High

68.8(67.7-69.8)

15.0(14.2-15.9)

16.2(15.5-17.0)

46.4(45.8-46.9)

16.9(16.4-17.3)

36.8(36.2-37.3)

<0.001

No response

76.9(74.4-79.5)

10.9(8.9-12.9)

12.1(10.2-14.1)

52.2(51.0-53.3)

20.3(19.3-21.2)

27.6(26.5-28.6)

<0.001

# standardized by age, gender and region based on the 2010 national population; ## sample weighted results; * Active transportation: walking or cycling; ** Public transportation: taking a bus/subway/shuttle bus; *** Inactive transportation: driving or taking a car/motorcycle/taxi/an electric bicycle; p Value is the test result between two different survey.

(5)3.3. Association between Tranportation Modes and Socio-demographic Characteristics (page 6,7 line 180-197)

We entered socio-demographic characteristic factors into multinomial logistic regression model to evaluate the association with transportation modes. Table 3 summarized the factors associated with active transportation and public transportation. Using the inactive transportation as a reference, all included socio-demographic characteristic factors significantly affected the other two classes (p<0.001). In both surveys, female was significantly more likely to use active transportation (OR (95% CI): 4.41(4.14-4.70), 2.50(2.44-2.57) respectively for 2002 and 2010-2012, where the male was reference) and public transportation (OR (95% CI): 3.68(3.36-4.03), 2.20(2.12-2.28) respectively for 2002 and 2010-2012, where the male was reference). Whereas participants with high annual average income per capita were less likely to use active transportation and public transportation. The prevalence of active transportation and public transportation increased with the age. The higher the education level the higher the prevalence of public transportation. Using active transportation was significantly more common in participants living in rural areas than those living in urban areas in 2002. Public transportation use was more pronounced in urban areas than rural areas for both surveys, however, the gap was sharply reduced in 2010-2012.

Table 3. Survey multinomial logistic regress analysis of factors associated with transportation modes #

Characteristics

2002 (OR(95%CI))##

2010-2012(OR(95%CI)) ##

Active*

Public**

Active*

Public**

Gender

Male

1.0

1.0

1.0

1.0

Female

4.41(4.14-4.70)

3.68(3.36-4.03)

2.50(2.44-2.57)

2.20(2.12-2.28)

Age group

15-29.9

1.0

1.0

1.0

1.0

30-44.9

0.84(0.79-0.89)

0.42(0.38-0.47)

0.55(0.53-0.58)

0.37(0.35-0.39)

45-55.9

2.05(1.89-2.22)

0.83(0.74-0.93)

1.10(1.06-1.14)

0.51(0.48-0.53)

≥60

10.18(8.35-12.42)

2.08(1.62-2.68)

3.51(3.34-3.69)

1.16(1.09-1.23)

Region type

Urban areas

1.0

1.0

1.0

1.0

Rural areas

1.56(1.47-1.65)

0.16(0.14-0.17)

0.68(0.66-0.70)

0.41(0.40-0.43)

Education level

Primary school or below

1.0

1.0

1.0

1.0

Junior or Senior high school

0.54(0.50-0.58)

1.64(1.42-1.90)

0.74(0.72-0.77)

1.03(0.99-1.08)

College and above

0.60(0.53-0.67)

3.09(2.58-3.71)

0.61(0.57-0.64)

1.62(1.51-1.73)

Annual average income per capita

Low

1.0

1.0

1.0

1.0

Middle

0.45(0.42-0.48)

0.86(0.77-0.95)

0.73(0.71-0.76)

0.92(0.88-0.96)

High

0.22(0.20-0.23)

0.63(0.57-0.71)

0.55(0.53-0.57)

0.78(0.75-0.82)

No response

0.38(0.32-0.45)

0.73(0.57-0.94)

0.77(0.73-0.82)

1.00(0.93-1.07)

# standardized by age, gender and region based on the 2010 national population, inactive transportation was used as class of reference; ## sample weighted results; * Active transportation: walking or cycling; ** Public transportation: taking a bus/subway/shuttle bus.

(6)4.Discussion (page 9 line 246-247 )

From 2002 to 2010-2012, the proportion of public transportation increased from 7.5% to 15.7%, nevertheless, the increase was much lower than the inactive transportation.

(7)5.Conclusions (page 9 line 268,269)

In rural area, with higher education level and annual average income per capita were negatively with active transportation in 2010-2012.

Round 2

Reviewer 2 Report

Thank you for your thoughtful responses to my proposed edits. I think the findings are quite fascinating. 

Reviewer 3 Report

The authors have addressed my main concerns